# Lower Hippocampal Volume in Patients with Schizophrenia and Bipolar Disorder: A Quantitative MRI Study

**DOI:** 10.3390/jpm11020121

**Published:** 2021-02-13

**Authors:** Jinya Sato, Yoji Hirano, Noriaki Hirakawa, Junichi Takahashi, Naoya Oribe, Hironori Kuga, Itta Nakamura, Shogo Hirano, Takefumi Ueno, Osamu Togao, Akio Hiwatashi, Tomohiro Nakao, Toshiaki Onitsuka

**Affiliations:** 1Department of Neuropsychiatry, Graduate School of Medical Sciences, Kyushu University, Fukuoka 812-8582, Japan; jinya.sato56@gmail.com (J.S.); hirakawa34@gmail.com (N.H.); yoonil1977@gmail.com (J.T.); n.oribe@mac.com (N.O.); hirokuga.370@gmail.com (H.K.); nakamura.itta.352@m.kyushu-u.ac.jp (I.N.); hiranoshogo@gmail.com (S.H.); nakao.tomohiro.275@m.kyushu-u.ac.jp (T.N.); 2Institute of Industrial Science, The University of Tokyo, Tokyo 153-8505, Japan; 3Hizen Psychiatric Medical Center, Division of Clinical Research, National Hospital Organization, Saga 842-0192, Japan; uenotk@gmail.com; 4Department of Molecular Imaging and Diagnosis, Graduate School of Medical Sciences, Kyushu University, Fukuoka 812-8582, Japan; togao@radiol.med.kyushu-u.ac.jp; 5Department of Clinical Radiology, Graduate School of Medical Sciences, Kyushu University, Fukuoka 812-8582, Japan; hiwatasi@radiol.med.kyushu-u.ac.jp

**Keywords:** schizophrenia, bipolar disorder, MRI, hippocampus, volume reduction

## Abstract

Since patients with schizophrenia (SZ) and bipolar disorder (BD) share many biological features, detecting biomarkers that differentiate SZ and BD patients is crucial for optimized treatments. High-resolution magnetic resonance imaging (MRI) is suitable for detecting subtle brain structural differences in patients with psychiatric disorders. In the present study, we adopted a neuroanatomically defined and manually delineated region of interest (ROI) method to evaluate the amygdalae, hippocampi, Heschl’s gyrus (HG), and planum temporale (PT), because these regions are crucial in the development of SZ and BD. ROI volumes were measured using high resolution MRI in 31 healthy subjects (HS), 23 SZ patients, and 21 BD patients. Right hippocampal volumes differed significantly among groups (HS > BD > SZ), whereas left hippocampal volumes were lower in SZ patients than in HS and BD patients (HS = BD > SZ). Volumes of the amygdalae, HG, and PT did not differ among the three groups. For clinical correlations, there were no significant associations between ROI volumes and demographics/clinical symptoms. Our study revealed significant lower hippocampal volume in patients with SZ and BD, and we suggest that the right hippocampal volume is a potential biomarker for differentiation between SZ and BD.

## 1. Introduction

Patients with schizophrenia (SZ) and bipolar disorder (BD) share many biological features, including genetic architecture (e.g., [1,2]) and structural (e.g., [3]) and functional brain abnormalities (e.g., [4,5,6,7,8,9,10]). Clinically, it is an urgent issue to accurately diagnose and distinguish between the two diseases for providing appropriate treatment and improving patient outcome, as they sometimes present similar pathologies. Thus, the identification of anatomical and functional biomarkers is crucial for differentiating SZ and BD patients to optimize pharmacological and psychosocial treatments for each condition [11]. Subcortical and cortical abnormalities in psychoses have been reported extensively, and high-resolution magnetic resonance imaging (MRI) is well-suited for detecting subtle brain structural differences in these populations [12,13,14,15]. A gold standard approach in MRI research is the use of neuroanatomically defined and manually delineated regions of interest (ROIs). A voxel-based morphometric (VBM) method is useful to investigate the whole brain, but the anatomical warping and density-based methodology can lead to failure in detecting abnormalities that are identified using ROI analyses [16]. Furthermore, it has been suggested that the analytical flexibility and the complex and variable workflows have substantial effects on results [17]; therefore, simpler MRI analysis methods still carry value. In the present study, we adopted a neuroanatomically defined and manually delineated ROI method to evaluate the amygdalae, hippocampi, Heschl’s gyrus (HG), and planum temporale (PT) as regions that have been shown to be crucial in the development of SZ and BD [18,19,20,21].

Medial temporal lobe changes are one of the most consistent findings in MRI studies of SZ and BD. The hippocampus is involved in memory and spatial learning [22]. Hippocampal volume reductions have been reported repeatedly in patients with SZ and BD [23]. The amygdala is an almond-shaped nucleus located deep within the medial temporal lobes and plays a primary role in the processing and memory of emotional reactions [24,25]. Both increases and decreases in amygdala volumes have been reported in patients with BD, although a recent meta-analysis found no significant amygdala volume changes in BD [26]. Conversely, an international collaboration for the study of BD, the Enhancing Neuroimaging Genetics through Meta-Analysis (ENIGMA) Consortium, reported significantly smaller amygdaloid volumes in BD compared with healthy subjects (HS) [27].

The HG includes the primary auditory cortex [28], and the left PT plays an important role in language processing [29]. Structural abnormalities of HG and PT [18,30], alongside auditory and language-related deficits [31,32], are observed frequently in patients with SZ. For example, patients with first-episode SZ showed lower left PT and bilateral HG gray matter volumes compared with HS and patients with BD [30]. In addition, Kasai et al. reported progressive volume reductions in the left HG and PT gray matter in first-episode SZ patients, which were not observed in first-episode affective psychosis patients or HS [18]. More recently, Hirano et al. investigated the association between neurophysiological and structural deficits in SZ patients and found spontaneous gamma oscillation deficits [33] and cortical volume deficits in HG [34]. In BD patients, Reite et al. reported alterations in structural/functional asymmetry in HG [18]. Moreover, Takahashi et al. reported that BD patients showed significantly lower left PT volumes compared with HS [35].

In the present study, we adopted a neuroanatomically defined and manually delineated ROI method to evaluate the amygdalae, hippocampi, HG, and PT. As noted before, functional or structural abnormalities have been reported in both SZ and BD in these four ROIs. We hypothesized that medial temporal lobe abnormalities (mainly hippocampus) may be common to both SZ and BD, and lateral temporal lobe abnormalities may be specific to SZ.

## 2. Materials and Methods

### 2.1. Participants

We analyzed magnetic resonance (MR) images of 23 patients with SZ, 21 patients with BD, and 31 HS. All subjects had normal hearing, were aged 20–63 years, and were right-handed [36]. After a detailed description of the study, all participants provided informed consent, according to the regulations of the Ethics Committee of the Graduate School of Medical Sciences, Kyushu University. Experimental procedures were approved by the Kyushu University Institutional Review Board for Clinical Trials (approval number: 26018, date: June 12, 2014), and conformed to the Declaration of Helsinki. The exclusion criteria were (1) neurological illness or major head trauma, (2) having received electroconvulsive therapy, (3) alcohol or drug dependence, (4) alcohol or drug abuse within the past 5 years, or (5) a verbal intelligence quotient below 75. HS were screened using the Structured Clinical Interview nonpatient edition. None of the HS or their first-degree relatives had an Axis-I psychiatric disorder. The socioeconomic status (SES) of the subjects and their parental SES were measured using the Hollingshead two-factor index [37]. For SZ, the Positive and Negative Syndrome Scale (PANSS) [38] was administered to assess the severity of psychiatric symptoms. The Young Mania Rating Scale (YMRS) [39] and the Hamilton Depression Rating Scale (HAM-D) [40] were administered to assess the severity of mood symptoms in BD. Table 1 shows the demographic and clinical characteristics of the participants.

### 2.2. MRI Data Acquisition

T1-weighted MR images were acquired using a three-dimensional (3D) turbo field echo sequence on a 3 Tesla scanner (Achieva TX, Philips Healthcare, Best, The Netherlands) at the Department of Radiology, Kyushu University Hospital. The imaging variables were as follows: repetition time = 8.2 ms, echo time = 3.8 ms, flip angle = 8°, field of view = 24 × 24 cm, number of echoes = 1, matrix = 240 × 240, inversion time = 1025.9 ms, number of slices = 190, and slice thickness = 1 mm. Images were aligned using the anterior and posterior commissure line and the sagittal sulcus to correct head tilt.

Total gray matter, white matter, and cerebrospinal fluid (CSF) volumes were calculated using Statistical Parametric Mapping 12 (SPM12, http://www.fil.ion.ucl.ac.uk/spm (accessed on 7 October 2020)), and VBM was performed using SPM12, running in MATLAB R2014a (The Math Works Inc., Natick, MA, USA). T1-weighted images were first segmented into gray matter, white matter, and CSF sections using tissue probability maps based on the International Consortium of Brain Mapping template for East Asian brains. Subsequently, we performed diffeomorphic anatomical registration through exponentiated lie algebra in SPM12 for intersubject registration of gray matter images [41]. The registered images were then smoothed with a Gaussian kernel of 8 mm full-width half-maximum and transformed into Montreal Neurological Institute stereotactic space using affine and nonlinear spatial normalization, implemented in SPM12. Total gray matter, white matter, and CSF volumes were generated from the VBM analysis. Total intracranial volume (ICV) was calculated as the sum of the gray matter, white matter, and CSF volumes.

### 2.3. Regions of Interest and Volume Measurement

The gray matter of the HG and PT, hippocampi, and amygdalae were outlined manually on a PC without knowledge of diagnosis (Figure 1). We used 3D information to provide reliable measures of the ROIs using a software package for medical image analysis [3D slicer, http://www.slicer.org (accessed on 7 October 2020)].

We used similar criteria to the previous works of Kwon et al. and Barta et al. for delineating HG and PT [42,43]. Briefly, HG was first identified in the most posterior coronal slice. Investigators manually drew the gray matter of HG from the most posterior to the most anterior slice. The most anterior slice of HG was defined as the last slice that HG could be identified anteriorly. In most cases, HG represented a single transverse convolution. In cases of more than one transverse convolution, we followed the literature’s definition [44,45,46,47]: when multiple convolutions originated medially from a common stem, these were all were defined as HG; however, when they originated separately from the retroinsular region, only the most anterior gyrus was labeled as HG, and the more posterior gyri were identified as PT. The anterior border of PT was defined by the posterior border of HG. Posteriorly, investigators traced the gray matter of the PT on the coronal images to the end of the Sylvian fissure. The amygdalae and hippocampi were also outlined manually. For the amygdala and hippocampus, the most anterior slice used for the amygdala was the slice in which the white matter tract linking the temporal lobe with the rest of the brain (temporal stem) could be seen. The most posterior slice of the hippocampus was the last appearance of fibers of the crux of the fornix. Once drawn, ROIs could be viewed in any plane as 3D objects for further editing. All editing was performed blind to diagnosis. Interrater reliability of the ROIs was evaluated by three independent raters (J.S., N.H., and J.T.), who were also blind to diagnosis. Five cases were selected at random, and every slice was edited by the raters. The intraclass correlations for interrater reliability were calculated.

### 2.4. Statistical Analysis

To correct for brain size differences, we used relative volumes, which were calculated using the following formula: relative volume (%) = (absolute ROI volume/ICV) × 100. For the ROI analysis of the subcortical structures, we used a mixed model repeated measures analysis of variance (ANCOVA) with group (SZ, BD, and HS) as the between-subjects factor, hemisphere (left or right) and region (amygdala or hippocampus) as the within-subjects factors, and age as a covariate. When significant interactions related to group were observed, we evaluated group differences separately for each region to identify the source of the interactions. We used follow-up ANCOVAs with group as the between-subjects factor, hemisphere as the within-subjects factor, and age as a covariate for each region. We performed one-way analyses of variance (ANOVAs) of the average volume across both hemispheres for each ROI. For the ROI analysis of the cortical structures, we used a mixed model ANCOVA with group (SZ, BD, and HS) as the between-subjects factor, hemisphere (left or right) and region (HG or PT) as the within-subjects factors, and age as a covariate. Post hoc tests were the same as the subcortical structure analyses. All statistical tests in ANCOVA and ANOVA were 2-tailed with *alpha* = 0.05. Exploratory analyses of the relationship between relative volumes of the ROIs and psychopathology scales were evaluated using *Spearman’s rho*. Due to our analyses involving multiple correlations, we set alpha levels at a conservative *p* < 0.001.

When significant group differences were observed in a particular region, optimal sensitivity and specificity of the ROI for the diagnosis of HS, SZ, or BD were determined via receiver operating characteristic (ROC) curve analysis using a nonparametric approach. We calculated the Youden index for each cutoff value as corresponding ((sensitivity + specificity) − 1) to find the cutoff values that maximized discriminating power. We used IBM SPSS Statistics 26 for the analyses.

## 3. Results

### 3.1. Reliability

The intraclass correlations for interrater reliability were 0.95 for the left hippocampus, 0.92 for the right hippocampus, 0.91 for the left amygdala, 0.90 for the right amygdala, 0.91 for left HG, 0.93 for right HG, 0.91 for the left PT, and 0.95 for the right PT.

### 3.2. Relative Volumes of Each ROI

Table 2 shows the relative volumes of each ROI. For the amygdalae and hippocampi, the repeated measures ANCOVA showed significant hemisphere-by-region-by-group (*F*[2,71] = 3.56, *p* = 0.03) and region-by-group interactions (*F*[2,71] = 8.72, *p* < 0.0001) and significant main effects of group (*F*[2,71] = 6.25, *p* = 0.003) and region (*F*[1,71] = 102.23, *p* < 0.0001) but not hemisphere (*F*[1,71] = 1.17, *p* = 0.28). There was no significant hemisphere-by-group (*F*[2,71] = 1.38, *p* = 0.26) or hemisphere-by-region (*F*[1,71] = 0.04, *p* = 0.84) interactions. To delineate the significant hemisphere-by-region-by-group interactions, post hoc tests were performed for each ROI.

For the amygdalae, the repeated-measures ANCOVA showed no significant main effects of group (*F*[2,71] = 0.53, *p* = 0.59) or hemisphere (*F*[2,71] = 1.15, *p* = 0.29), or a significant hemisphere-by-group interaction (*F*[2,71] = 0.39, *p* = 0.68). These results indicated that there were no significant group differences among the three groups in either hemisphere.

For the hippocampus, the repeated measures ANCOVA revealed a significant hemisphere-by-group interaction (*F*[2,71] = 3.56, *p* = 0.03) and a main effect of group (*F*[2,71] = 9.21, *p* < 0.0001) but not hemisphere (*F*[1,71] = 0.32, *p* = 0.57). In the right hemisphere, the one-way ANOVA showed a significant group difference (*F*[2,72] = 15.06, *p* < 0.0001), and post hoc Tukey’s honestly significant difference (HSD) tests revealed BD < HS (*p* = 0.04), SZ < HS (*p* < 0.0001) and SZ < BD (*p* = 0.03) right hippocampal volumes. These results indicated that right hippocampal volumes were highest in HS, lower in the BD patients, and lowest in the SZ patients. In the left hemisphere, the one-way ANOVA showed a significant group difference (*F*[2,72] = 7.47, *p* = 0.001), and post hoc HSD tests revealed SZ < HS (*p* = 0.001) and SZ < BD (*p* = 0.04) hippocampal volumes. No significant difference was found for the left hippocampus between HS and BD patients (*p* = 0.58). These results revealed that SZ patients had a lower left hippocampal volume compared with HS and BD patients (see Figure 2).

For HG and PT, the repeated-measures ANCOVA showed a significant main effect of region (*F*[1,71] = 20.85, *p* < 0.001). There were no other significant main effects or interactions, which indicated no significant group differences in the HG or PT (see Figure 3).

We also performed repeated-measures ANCOVAs using age and sex as covariates for ROI analysis; statistical results were the same using both methods.

### 3.3. Correlations

Regarding the associations between ROI volume and demographic/clinical variables, we found no significant correlations in HS (−0.34 ≤ *rho* ≤ 0.38, 0.04 ≤ *p* ≤ 0.89), SZ patients (−0.34 ≤ *rho* ≤ 0.43, 0.04 ≤ *p* ≤ 0.98), or BD patients (−0.40 ≤ *rho* ≤ 0.48; 0.03 ≤ *p* ≤ 0.94).

### 3.4. ROC Analysis

Since significant differences among three groups were observed in the right hippocampus, we used ROC curve analysis to explore the discriminatory value of the relative volumes. Figure 4 shows the ROC curve of relative volumes of the right hippocampus between HS and BD, HS and SZ, and BD and SZ. The area under the curve (AUC) of the ROC analysis in BD vs. SZ was maximal for the right hippocampal relative volume (AUC = 0.70, Standard error = 0.08, *p* = 0.03, 95% CI = 0.54–0.85), indicating that the right hippocampal volume could be used to differentiate between BD and SZ subjects with moderate accuracy. The Youden index indicated a favorable cutoff point of 0.20%, which resulted in 62% sensitivity and 74% specificity.

## 4. Discussion

We investigated the volumes of the hippocampi, amygdalae, HG, and PT in SZ patients, BD patients, and HS. We found that right hippocampal volumes were lower in BD patients compared with controls, and further lower in SZ patients, whereas left hippocampal volumes were lower exclusively in SZ patients. There were no significant differences in either amygdala, HG, or PT among the three groups.

Our results are consistent with many previous studies (e.g., [27,48]) although the recent study [14] of the direct comparison found no significant volume differences between the two groups. Our findings suggest that the lateralized hippocampal volume abnormality is related to the development of SZ and BD. We also found that the lower left hippocampal volume was not observed in BD patients and was specific to SZ, which is in line with numerous studies that have reported left hemisphere dominant abnormalities in SZ [49].

Although the degree of lowered volume in BD patients was not to that observed in SZ patients, the volume of the right hippocampus was found to be significantly lower in BD compared with HS. The ENIGMA Consortium reported significant lower volumes in the hippocampus, thalamus, and amygdala in patients with BD [27]; however, they did not evaluate laterality. We revealed that lower right hippocampal volume may be associated with pathology of BD. We highlight that in addition to large sample automated analysis methods, alternative approaches, such as the manual ROI drawing method, are important and valuable.

In the present study, there were no significant differences in either amygdala among the three groups. Moreover, contrary to our hypotheses and findings of previous studies (e.g., [18]), we did not find significant group differences in HG or PT. Bryant et al. reported no differences in the volumes of temporal lobe structures between female SZ patients and HS [50]. In addition, Bora et al. performed voxelwise meta-analysis and reported that gray matter reductions of SZ and BD were less severe in sex-balanced samples [15]. Therefore, sex effects should be considered in future studies in larger sample sizes. We did not find any significant associations between the volumes of the ROIs and demographics/clinical symptoms. Although hippocampal volume reductions may increase the risk of psychosis, they may not be directly related to the severity of symptoms. Given the function of the hippocampus, it will be important to investigate relationships between hippocampal volumes and cognitive functional domains in future studies.

Because SZ and BD share the biological basis of psychoses [1,2,3,4,5,6,7,8,9,10,11,51,52,53,54,55,56,57] and they sometimes present similar pathologies, detection of anatomical and functional biomarkers that differentiate SZ and BD patients is crucial for the optimization of pharmacological and psychosocial treatments for each condition. The results of our ROC analyses suggest that the right hippocampal volume is a potential marker for differentiation between SZ and BD. To this end, multimodal functional neuroimaging studies [26,51,52,53,54,55,56,57,58,59,60,61] that combine structural MRI [3,16,18,19,20,21,23,26,27,28,42,43,44,51,61,62,63], functional MRI [26,53,54,55,56,57,58,59,60,61,64,65], magnetic resonance spectroscopy [66,67,68,69], positron emission tomography [69,70,71,72], electroencephalography, and magnetoencephalography [4,5,6,7,8,9,59,73,74,75,76,77,78,79,80,81,82,83,84,85] may lead to a better understanding of the biological bases of SZ and BD and enable the development of optimized treatments (e.g., [53,54,58,61,86,87]).

We must note that this study has some limitations. First, we had a relatively small sample size owing to our type of dataset and manual drawing approach, which requires considerable time and labor. Therefore, weaker correlations that were hypothesized may have gone undetected. Second, as mentioned earlier, sex differences were not explored in our study but warrant investigation in a larger sample, as suggested by Bryant et al. [50], Egloff et al. [88], and Bora et al. [15]. Third, the effect of the duration of illness should be considered. The current sample of BD had higher age and lower duration of illness compared to SZ. Thus, in the present study, we could not exclude the effect of earlier disease-specific differences and late unspecific processes, including disease progress or effects of medications. Importantly, metaregression analysis by Fuar-Poli et al. [89] reported that the higher the cumulative exposure to antipsychotic treatment, the greater the gray matter decreases in the SZ group over follow-up time, while no significant effects were observed for duration of illness. Although there were no significant correlations between any ROI volumes and duration of illness in our study, associations between the effect of disease duration and the hippocampal volumes should be examined in a future study. Finally, we could not exclude the medication effects on hippocampal volumes in clinical groups, as pointed out by a previous study [90].

In summary, we showed that patients with SZ and BD have significantly lower hippocampal volumes compared with HS. Furthermore, we suggest that the right hippocampal volume is a potential biomarker for differentiation between SZ and BD.

## Figures and Tables

**Figure 1 jpm-11-00121-f001:**
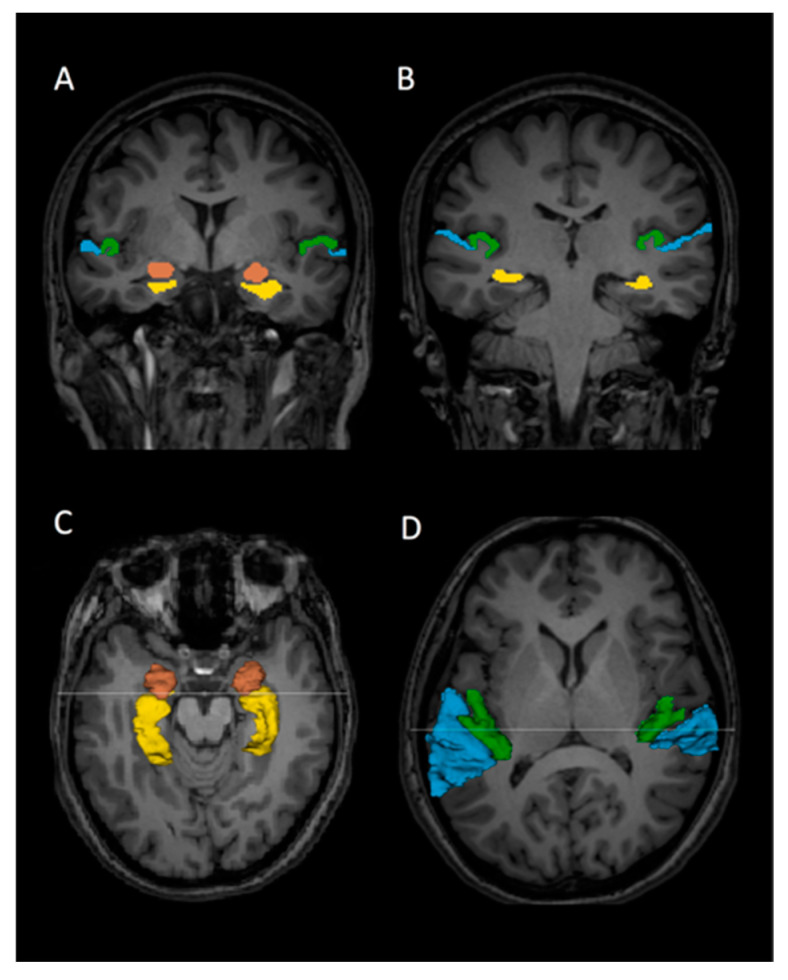
Delineation of Heschl’s gyrus (HG), planum temporale (PT), hippocampi, and amygdalae on 3 Tesla magnetic resonance imaging slices in a healthy subject. The HG (green), PT (blue), hippocampus (yellow), and amygdala (brown) regions of interest are outlined. (**A**,**B**): Coronal view of the HG, PT, hippocampi, and amygdalae. (**C**): Three-dimensional (3D) reconstruction of the hippocampi and amygdalae superimposed onto an axial slice. (**D**): 3D reconstruction of HG and PT gray matter superimposed onto an axial slice. The lines on **C** and **D** correspond to the coronal slices of **A** and **B**, respectively.

**Figure 2 jpm-11-00121-f002:**
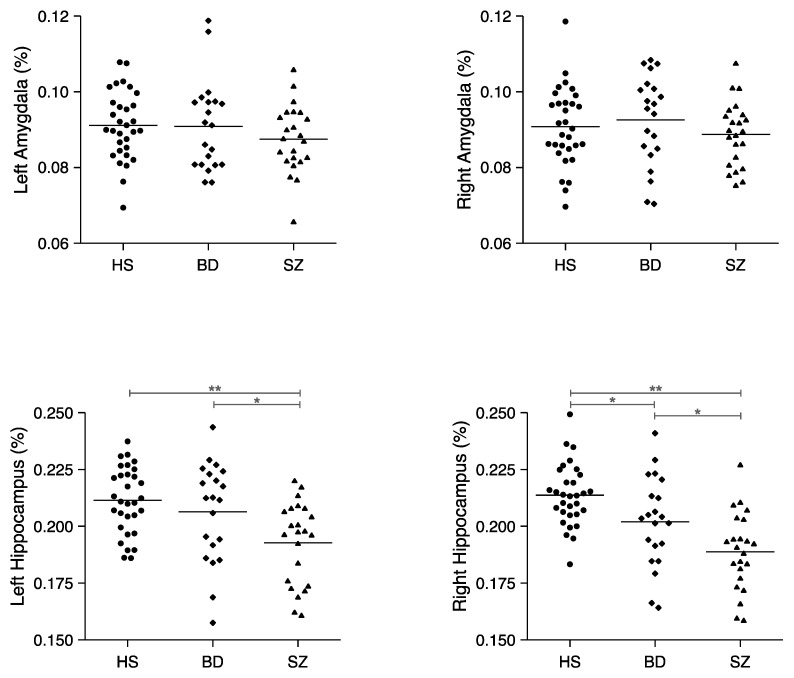
Relative volumes of the left and right amygdalae and hippocampi in healthy subjects (HS) (*n* = 31), patients with bipolar disorder (BD) (*n* = 21), and patients with schizophrenia (SZ) (*n* = 23). Horizontal lines indicate means. * *p* < 0.05, ** *p* < 0.001.

**Figure 3 jpm-11-00121-f003:**
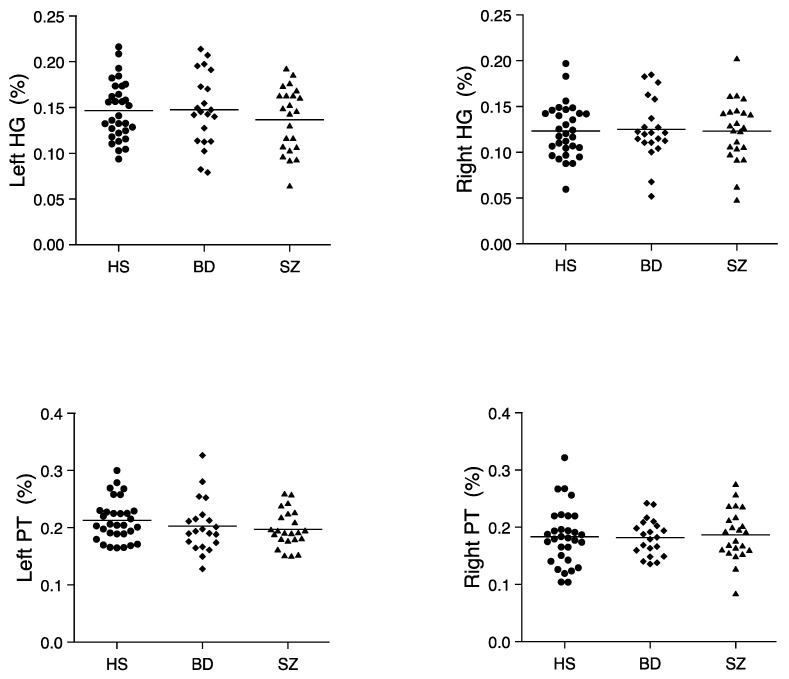
Relative volumes of the left and right HG and PT in HS (*n* = 31), BD (*n* = 21), and SZ (*n* = 23). Horizontal lines indicate means.

**Figure 4 jpm-11-00121-f004:**
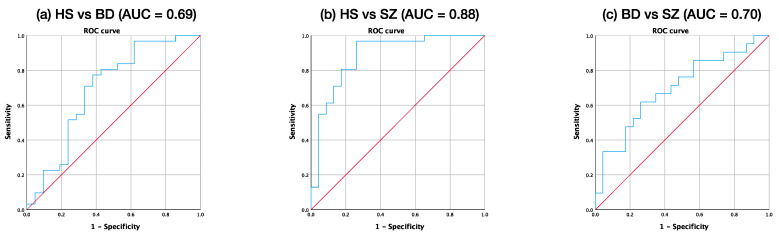
Receiver operating characteristic (ROC) curve of relative volumes of the right hippocampi between (**a**) HS and BD, (**b**) HS and SZ, and (**c**) BD and SZ. The area under the ROC curve was 0.69, 0.88, and 0.70, respectively.

**Table 1 jpm-11-00121-t001:** Demographic and clinical characteristics of the participants.

	Healthy Subjects (HS)	Patients with Bipolar Disorder (BD)	Patients with Schizophrenia (SZ)	*χ*2 or *F* or *t*	*df*	*p*
Male/Female	11/20	11/10	8/15	1.66	2	0.44
Age (years)	49.2 ± 9.8	48.9 ± 9.1	44.3 ± 6.2	2.50	2, 73	0.09
Handedness ^a^	92.0 ± 32.7	82.4 ± 51.0	96.7 ± 8.7	0.99	2, 73	0.38
SES ^b^	2.4 ± 0.8	2.9 ± 1.0	3.6 ± 1.0	10.66	2, 45	<0.001
Parental SES	3.1 ± 0.9	2.9 ± 0.9	2.8 ± 0.9	1.10	2, 45	0.34
Education (years)	14.4 ± 2.6	14.2 ± 2.3	14.4 ± 2.0	0.95	2, 45	0.39
Symptom onset ^#^ (years)		37.8 ± 11.2	26.7 ± 6.7	31.95	42	<0.001
Medication dose (CPZ equiv., mg)		144.2 ± 183.9	599.8 ± 328.4	5.60	42	<0.001
PANSS positive			18.1 ± 7.9			
PANSS negative			22.5 ± 7.7			
PANSS general			42.9 ± 15.2			
YMRS		5.2 ± 5.7				
SIGH-D score		10.3 ± 5.9				

Values are means ± SD unless otherwise noted. ^a^ Edinburgh Handedness Inventory, ^b^ Hollingshead two factor Index of Social Position. SES = socioeconomic status, YMRS = Young Mania Rating Scale, SIGH-D = Structured Interview Guide for the Hamilton Depression Rating Scale. ^#^ Symptom onset is the age at which the symptoms appeared.

**Table 2 jpm-11-00121-t002:** Volume of each region of interest (ROI).

	Healthy Subjects (HS)	Patients with Bipolar Disorder (BD)	Patients with Schizophrenia (SZ)	*df*	*F*	*p*
Intracranial Volume (mL)	1380 ± 128	1422 ± 136	1418 ± 157	2, 72	0.79	0.46
Left Hippocampus						
Absolute (mL)	2.91 ± 0.22	2.92 ± 0.24	2.72 ± 0.24			
Relative (%)	0.211 ± 0.015	0.206 ± 0.022	0.193 ± 0.018	2, 72	7.47	0.001 *
Right Hippocampus						
Absolute (mL)	2.95 ± 0.29	2.86 ± 0.20	2.67 ± 0.28			
Relative (%)	0.214 ± 0.014	0.202 ± 0.020	0.189 ± 0.017	2, 72	15.06	<0.0001 ^†^
Left Amygdala						
Absolute (mL)	1.26 ± 0.17	1.28 ± 0.14	1.24 ± 0.16			
Relative (%)	0.091 ± 0.009	0.091 ± 0.012	0.087 ± 0.009	2, 72	1.01	0.37
Right Amygdala						
Absolute (mL)	1.25 ± 0.20	1.31 ± 0.14	1.26 ± 0.17			
Relative (%)	0.091 ± 0.010	0.093 ± 0.012	0.089 ± 0.008	2, 72	0.78	0.46
Left Heschl’s gyrus						
Absolute (mL)	2.03 ± 0.50	2.02 ± 0.53	1.95 ± 0.59			
Relative (%)	0.147 ± 0.032	0.148 ± 0.039	0.137 ± 0.035	2, 72	0.71	0.50
Right Heschl’s gyrus						
Absolute (mL)	1.71 ± 0.48	1.77 ± 0.46	1.75 ± 0.55			
Relative (%)	0.123 ± 0.029	0.125 ± 0.034	0.123 ± 0.034	2, 72	0.03	0.97
Left Planum temporale						
Absolute (mL)	2.94 ± 0.60	2.88 ± 0.68	2.79 ± 0.56			
Relative (%)	0.213 ± 0.036	0.203 ± 0.046	0.197 ± 0.032	2, 72	1.20	0.31
Right Planum temporale						
Absolute (mL)	2.53 ± 0.76	2.56 ± 0.36	2.63 ± 0.65			
Relative (%)	0.183 ± 0.050	0.182 ± 0.032	0.186 ± 0.044	2, 72	0.06	0.94

* Post hoc tests indicated that schizophrenia (SZ) was significantly different from bipolar disorder (BD) and healthy subjects (HS) (Tukey’s honestly significant difference test, *p* < 0.05). ^†^ Post hoc tests indicated that volumes were significantly different as follows; HS > BD > SZ (Tukey Honestly Significant Difference, *p* < 0.05).

## Data Availability

The data that support the findings of this study are available on request from the corresponding authors upon reasonable request. The data are not publicly available due to privacy or ethical restrictions.

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
