# Peer review of "Lower Hippocampal Volume in Patients with Schizophrenia and Bipolar Disorder: A Quantitative MRI Study"

_jpm, 2021, doi:10.3390/jpm11020121_

Round 1

Reviewer 1 Report

In this work the authors investigated differences in the volume of the amygdalae, hippocampi, Heschl’s gyrus (HG) and planum temporale (PT), manually delineated from high-resolution MRI scans, between 23 schizophrenia (SZ) patients, 21 bipolar disorder (BD) patients and 31 healthy subjects (HS). They found decreased volume of the right hippocampus (SZ < BD < HS) and left hippocampus in SZ (SZ < BD = HS). These differences were not correlated with clinical symptoms.

General comment:

The field can benefit from direct comparisons of such hard to differentiate diseases. This works is relevant in that context. However, I am not convinced of the discriminative ability of the laterality of hippocampal volume reduction between clinical groups, which I would see as the added value of this study. Authors should show a sensitivity/specificity analysis (e.g. ROC curve) on this. Furthermore, there are several confounding issues in the analysis (see specific comments), which reduces the objectivity of the conclusions. They should be addressed before this manuscript can deemed publishable, in this reviewer's humble opinion.

Strengths:
1. It is a direct side-by-side comparison between two commonly overlapping diseases (SZ and BD).

2. Manual delineation of ROIs.

Limitations:
1. Why focusing only on these 4 ROIs and not other regions associated with SZ and BD in the literature, such as the thalamus.

2. Gender, disease duration and medication have well known effects on the volume of brain structures and do not seem to have been considered in the analysis. They could have been added as covariates to the statistical models.

Specific comments:

  1. Lines 45-47. Authors could cite previous studies of brain structure in these populations, e.g. (Brugger and Howes, 2017), (Ellison-Wright and Bullmore, 2010), and many other. Namely, the authors could place their study along with other studies directly comparing SZ and BD such as (Madeira et al., 2020) and (Bora et al., 2012).

Madeira, N., Duarte, J.V., Martins, R., Costa, G.N., Macedo, A., Castelo-Branco, M., 2020. Morphometry and gyrification in bipolar disorder and schizophrenia: a comparative MRI study. Neuroimage: Clinical. 102220. DOI: 10.1016/j.nicl.2020.102220.

Bora, E., Fornito, A., Yucel, M., Pantelis, C., 2012. The effects of gender on grey matter abnormalities in major psychoses: a comparative voxelwise meta-analysis of schizo- phrenia and bipolar disorder. Psychol. Med. 42, 295–307. https://doi.org/10.1017/ S0033291711001450.

Ellison-Wright, I., Bullmore, E., 2010. Anatomy of bipolar disorder and schizophrenia: a meta-analysis. Schizophr. Res. 117, 1–12. https://doi.org/10.1016/j.schres.2009.12. 022.

Brugger, S.P., Howes, O.D., 2017. Group heterogeneity and homogeneity of regional brain structure in schizophrenia: a meta-analysis. JAMA Psychiatry 74, 1104–1111. https://doi.org/10.1001/jamapsychiatry.2017.2663.Heterogeneity.

  1. There is a certain degree of repetition of the same ideas, namely in the introduction. E.g. the last paragraph is simply a repetition of previous sentences, adding no new information/objectives.

  1. There is an imbalance of gender distribution in the two clinical groups. While females are more in the SZ group, males are in higher number in the BD group. Gender differences in GM volumes are a known source of heterogeneity in comparisons between SZ and BD; when gender is controlled for, GM abnormalities are much more restricted.

Bora, E., Fornito, A., Yucel, M., Pantelis, C., 2012. The effects of gender on grey matter abnormalities in major psychoses: a comparative voxelwise meta-analysis of schizo- phrenia and bipolar disorder. Psychol. Med. 42, 295–307. https://doi.org/10.1017/ S0033291711001450.

  1. Included patients have a wide age range and a relatively high average age, furthermore with a high and different disease duration, according to age at symptom onset: ~10 years in BD and ~13-14 years in SZ. SZ patients are younger and reported symptoms much earlier than BD patients. There is conflicting evidence on the impact of disease duration on the volume of deep grey matter, with some studies showing no effect (Fusar-Poli et al., 2013) whilst others demonstrated a positive association with GM volumes (Hashimoto et al., 2018). Although causality is uncertain, it might be hypothesized that such morphological changes were acquired rather than reflecting a neurodevelopmental nature. The fact that in this study BD patients had higher age and lower disease duration compared to SZ patients might hinder the investigation of earlier disease-specific changes and measures might be potentially influenced by late unspecific processes. This should at least be discussed.

Fusar-Poli, P., Smieskova, R., Kempton, M.J., Ho, B.C., Andreasen, N.C., Borgwardt, S., 2013. Progressive brain changes in schizophrenia related to antipsychotic treatment? A meta-analysis of longitudinal MRI studies. Neurosci. Biobehav. Rev. 37, 1680–1691. https://doi.org/10.1016/j.neubiorev.2013.06.001.

Hashimoto, N., Ito, Y.M., Okada, N., Yamamori, H., Yasuda, Y., Fujimoto, M., Kudo, N., Takemura, A., Son, S., Narita, H., Yamamoto, M., Tha, K.K., Katsuki, A., Ohi, K., Yamashita, F., Koike, S., Takahashi, T., Nemoto, K., Fukunaga, M., Onitsuka, T., Watanabe, Y., Yamasue, H., Suzuki, M., Kasai, K., Kusumi, I., Hashimoto, R., 2018. The effect of duration of illness and antipsychotics on subcortical volumes in schi- zophrenia: analysis of 778 subjects. NeuroImage Clin. 17, 563–569. https://doi.org/ 10.1016/j.nicl.2017.11.004.

  1. I do not feel a sufficiently convincing explanation of why authors restricted the analysis to the 4 ROIs described in this manuscript. What is the importance of these regions more than others reported across many other studies of brain structural alteration in BD and SZ? Furthermore, while ROI studies have higher statistical power, VBM studies often replicate results of ROI studies and uncover significantly altered concentration of GM, due to higher sensitivity and exploratory power. The authors could at least discuss their results in light of other recent studies with a direct comparison of structural MRI signatures of SZ and BD (Madeira et al., 2020).

Madeira, N., Duarte, J.V., Martins, R., Costa, G.N., Macedo, A., Castelo-Branco, M., 2020. Morphometry and gyrification in bipolar disorder and schizophrenia: a comparative MRI study. Neuroimage: Clinical. 102220. DOI: 10.1016/j.nicl.2020.102220.

  1. “If significant group differences were observed in a particular region, we evaluated group differences separately for each region.” – how did authors identified differences between groups in certain regions, as the AN(C)OVA model gives the main effect of each factor, thus the effect of group across regions? This should be clarified.

“We used follow up ANCOVAs with group as the between-subjects factor, hemisphere as the within-subjects factor, and age as a covariate for each region. We performed one-way analyses of variance (ANOVAs) of the average volume across both hemispheres for each ROI.” – why these two analyses and not a single ANCOVA with merged hemisphere volumes? The rationale should be explained.

Why did authors analyzed subcortical and cortical regions separately and not together in a single mixed ANCOVA model?

  1. Authors could start the description of the results with significant differences in the hippocampi instead of non-significant results in the amygdalae.

  1. In Figure 2 authors could indicate statistically significant differences in the plots. Also, why show only amygdalae (non-significant) and not cortical regions or only hippocampi (significant)?

  1. I am not convinced of the discriminative ability of the laterality of hippocampal volume reduction between clinical groups, which I would see as the added value of this study. The differences in the left hippocampus follow the same trend as on the right, although not reaching significance between BD and SZ.

  2. The authors “suggest that the hippocampus is involved in the underlying pathophysiology of both SZ and BD” but they do not present ant explanation or supporting data for this claim. Even the correlations with clinical symptoms were not significant at all.

Author Response

General comments

  1. The field can benefit from direct comparisons of such hard to differentiate diseases. This works is relevant in that context. However, I am not convinced of the discriminative ability of the laterality of hippocampal volume reduction between clinical groups, which I would see as the added value of this study. Authors should show a sensitivity/specificity analysis (e.g. ROC curve) on this.

 As suggested, we have performed sensitivity/specificity analyses in the revised manuscript (MS). We have created Figure 4 to show ROC curves of relative volumes of the right hippocampus between healthy subjects (HS) and patients with bipolar disorder (BD), HS and patients with schizophrenia (SZ), BD and SZ. The area under the ROC curve was 0.69, 0.88 and 0.70 respectively. We have modified the Methods, Results and Discussion sections in the revised MS accordingly.

  1. Furthermore, there are several confounding issues in the analysis (see specific comments), which reduces the objectivity of the conclusions. They should be addressed before this manuscript can deemed publishable, in this reviewer's humble opinion.

We have performed repeated measures ANCOVAs with age and sex as covariates. In terms of amygdala and hippocampus, the repeated measures ANCOVA showed significant hemisphere-by-region-by-group (F [2,71] = 4.07, p = 0.02) and region-by-group interactions (F [2,71] =10.50, p < 0.0001) and significant main effects of group (F [2,71] = 6.86, p = 0.002) and region (F [1,71] = 134.46, p < 0.0001) with no significant main effect of hemisphere (F [1,71] = 0.86, p = 0.36) and no significant hemisphere-by-group (F [2,71] = 1.61, p = 0.21) and hemisphere-by-region (F [1,71] = 0.14, p = 0.71) interactions. In terms of HG and PT, the repeated measures ANCOVA showed significant main effect of region (F [1,71] =21.02, p<0.001) without other significant main effects and any significant interactions.

Thus, the results are the same using age and sex as covariates. We have clarified the results as follows: “We also performed repeated measure ANCOVAs using age and sex as covariates for ROI analysis; statistical results were the same using both methods.”

For disease duration and medication, it is not suitable to add these variables as covariates in the main ANCOVAs, because these variables do not apply to healthy subjects. Moreover, some SZ received mood stabilizer or benzodiazepine, and some BD subjects received mood stabilizer, benzodiazepine or antidepressant. Therefore, it is difficult to evaluate medication effect in the current results. For disease duration, there were no significant correlations between any ROI volumes and duration of illness in the present study. Therefore, we have discussed issues for disease duration and medication as the limitations of this study in the Discussion section as follows:

“Third, the effect of duration of illness should be considered. The current sample of BD had higher age and lower duration of illness compared to SZ. Thus, in the present study, we could not exclude the effect of earlier disease-specific differences and late unspecific processes including disease progress or effects of medications. Importantly, meta-regression analysis by Fuar-Poli et al. (2013) reported that the higher the cumulative exposure to antipsychotic treatment the greater the gray matter decreases in the SZ group over follow-up time, while no significant effects were observed for duration of illness. Although there were no significant correlations between any ROI volumes and duration of illness in our study, associations between the effect of disease duration and the hippocampal volumes should be examined in the future study. Finally, we could not exclude the medication effects on hippocampal volumes in clinical groups, as pointed out by a previous study (Hashimoto et al 2018).”

  1. Why focusing only on these 4 ROIs and not other regions associated with SZ and BD in the literature, such as the thalamus.

 As noted in the introduction section, functional or structural abnormalities have been reported in both SZ and BD in these 4 ROIs. We hypothesized that medial temporal lobe abnormalities (mainly hippocampus) may be common to both SZ and BD, and lateral temporal lobe abnormalities may be specific to SZ. In addition, prior to developing the plan of this study, we had experiences for manual delineation of these 4 regions reliably (TO, YH, and NO). Therefore, we chose these 4 regions as the current ROI analysis.

In the revised MS, we have clarified the motivation for investigating these 4 regions, where we now say: “As noted before, functional or structural abnormalities have been reported in both SZ and BD in these 4 ROIs. We hypothesized that medial temporal lobe abnormalities (mainly hippocampus) may be common to both SZ and BD, and lateral temporal lobe abnormalities may be specific to SZ.”

  1. Gender, disease duration and medication have well known effects on the volume of brain structures and do not seem to have been considered in the analysis. They could have been added as covariates to the statistical models.

Please see our response to reviewer #1’s general comment 2.

Specific comments

  1. Lines 45-47. Authors could cite previous studies of brain structure in these populations, e.g. (Brugger and Howes, 2017), (Ellison-Wright and Bullmore, 2010), and many other. Namely, the authors could place their study along with other studies directly comparing SZ and BD such as (Madeira et al., 2020) and (Bora et al., 2012).

 We greatly appreciate your comment. We have cited the papers in the revised MS.

  1. There is a certain degree of repetition of the same ideas, namely in the introduction. e.g. the last paragraph is simply a repetition of previous sentences, adding no new information/objectives.

We recognize that we were redundant. We have modified the last paragraph and clarified the motivation for the present study.

  1. There is an imbalance of gender distribution in the two clinical groups. While females are more in the SZ group, males are in higher number in the BD group. Gender differences in GM volumes are a known source of heterogeneity in comparisons between SZ and BD; when gender is controlled for, GM abnormalities are much more restricted.

We appreciate your comment. For the issue of ANCOVAs of the sex factor, please see our response to reviewer #1’s general comment 2. We have cited the paper and modified the discussion section as follows:

“In addition, Bora et al. performed voxelwise meta-analysis and reported that gray matter reductions of SZ and BD were less severe in sex-balanced samples.”

  1. Included patients have a wide age range and a relatively high average age, furthermore with a high and different disease duration, according to age at symptom onset: ~10 years in BD and ~13-14 years in SZ. SZ patients are younger and reported symptoms much earlier than BD patients. There is conflicting evidence on the impact of disease duration on the volume of deep grey matter, with some studies showing no effect (Fusar-Poli et al., 2013) whilst others demonstrated a positive association with GM volumes (Hashimoto et al., 2018). Although causality is uncertain, it might be hypothesized that such morphological changes were acquired rather than reflecting a neurodevelopmental nature. The fact that in this study BD patients had higher age and lower disease duration compared to SZ patients might hinder the investigation of earlier disease-specific changes and measures might be potentially influenced by late unspecific processes. This should at least be discussed.

The reviewer’s comments are reasonable. We have modified the discussion accordingly. Please see our response to reviewer #1’s general comment 2.

  1. I do not feel a sufficiently convincing explanation of why authors restricted the analysis to the 4 ROIs described in this manuscript. What is the importance of these regions more than others reported across many other studies of brain structural alteration in BD and SZ? Furthermore, while ROI studies have higher statistical power, VBM studies often replicate results of ROI studies and uncover significantly altered concentration of GM, due to higher sensitivity and exploratory power. The authors could at least discuss their results in light of other recent studies with a direct comparison of structural MRI signatures of SZ and BD (Madeira et al., 2020).

 We appreciate the reviewer’s comment. To clarify to investigate the 4 ROIs, we have clarified our hypotheses (please see our response to reviewer #1’s general comment 3). As suggested, we have modified the discussion section as follows:

“Our results are consistent with many previous studies [e.g., 27, 48] although the recent study [14] of the direct comparison found no significant volume differences between the two groups.”

  1. “If significant group differences were observed in a particular region, we evaluated group differences separately for each region.” – how did authors identified differences between groups in certain regions, as the AN(C)OVA model gives the main effect of each factor, thus the effect of group across regions? This should be clarified.

We recognize that we were unclear. We have modified the sentence as follows: “When significant interactions related to group were observed, we evaluated group differences separately for each region to identify the source of the interactions.”

“We used follow up ANCOVAs with group as the between-subjects factor, hemisphere as the within-subjects factor, and age as a covariate for each region. We performed one-way analyses of variance (ANOVAs) of the average volume across both hemispheres for each ROI.” – why these two analyses and not a single ANCOVA with merged hemisphere volumes? The rationale should be explained.

 We apologize for the error. We only used follow up ANCOVAs with group as the between-subjects factor, hemisphere as the within-subjects factor, and age as a covariate for each region. If significant group-by-hemisphere interactions were not observed, we performed one-way analyses of variance (ANOVAs) of the average volume across both hemispheres for each ROI.

Why did authors analyzed subcortical and cortical regions separately and not together in a single mixed ANCOVA model?

We hypothesized that medial temporal lobe abnormalities (mainly hippocampus) may be common to both SZ and BD, and lateral temporal lobe abnormalities may be specific to SZ. Each hypothesis is independent, and thus we analyzed subcortical and cortical regions separately.

  1. Authors could start the description of the results with significant differences in the hippocampi instead of non-significant results in the amygdalae.

As suggested, we have shown the results of hippocampi first.

  1. In Figure 2 authors could indicate statistically significant differences in the plots. Also, why show only amygdalae (non-significant) and not cortical regions or only hippocampi (significant)?

We have modified Figure 2 to indicate statistically significant differences. We have also added scatterplots of HG and PT in Figure 3.

  1. I am not convinced of the discriminative ability of the laterality of hippocampal volume reduction between clinical groups, which I would see as the added value of this study. The differences in the left hippocampus follow the same trend as on the right, although not reaching significance between BD and SZ.

 We appreciate your comment. We have performed sensitivity/specificity analyses in the revised MS (please see our response to your general comment 1). In addition, we have rephrased the conclusion as follows:

“Furthermore, we suggest that the right hippocampal volume is a potential biomarker for differentiation between SZ and BD.”

  1. The authors “suggest that the hippocampus is involved in the underlying pathophysiology of both SZ and BD” but they do not present ant explanation or supporting data for this claim. Even the correlations with clinical symptoms were not significant at all.

We agree. We have deleted the description of “suggest that the hippocampus is involved in the underlying pathophysiology of both SZ and BD” in the revised MS.

Reviewer 2 Report

In this manuscript Sato et al. study four temporal structures in order to detect structural differences in patients with schizophrenia, bipolar disorder and healthy subjects. Measures were done using manually segmented ROIs of amygdalae, hippocampi, Heschl’s gyrus, and planum temporale.

Authors found differences in right and left hippocampi but not in any other structure. There were no significant associations between ROI volumes and demographics/clinical symptoms.

How hippocampi volume differences between groups follow the trend HS > BD > SZ it is very interesting and confirms previous studies.

Major concerns:

1.-In Table1 it can be observe that there are significant differences in parental socioeconomic status, symptom onset and medication dose. This reviewer would like to see the analysis covariating for those variables.

2.-The authors could discuss deeply their hippocampi results.

Minor concerns:

1.-Is “symptom onset” the age at which the symptoms appear or the time that has happened since the symptoms appear? Please clarify.   

2.-Which statistical program was used.

Author Response

Major concerns:

In Table1 it can be observe that there are significant differences in parental socioeconomic status, symptom onset and medication dose. This reviewer would like to see the analysis covariating for those variables.

There was no significant group difference in parental socioeconomic status (p=0.34, please see Table 1).

For the issues of symptom onset (duration of illness) and medication dose, please see our response to reviewer #1’s general comment 2. We have discussed issues for disease duration and medication as the limitations of this study.

The authors could discuss deeply their hippocampi results.

We appreciate the reviewer’s comment. As pointed out by reviewer #1, we have performed sensitivity/specificity analyses for the right hippocampal volume. We have modified the MS according to reviewer #1’s comments to discuss hippocampi results further.

Minor concerns:

Is “symptom onset” the age at which the symptoms appear or the time that has happened since the symptoms appear? Please clarify.   

 “Symptom onset” is the age at which the symptoms appeared. We have clarified it in Table 1.

Which statistical program was used.           

We used IBM SPSS Statistics 26. We have clarified in the revised MS.

Round 2

Reviewer 1 Report

In this revision, the authors have undertaken efforts to address the points raised in the first review, including further data analysis. This effort is highly appreciated.

I thank the authors for addressing all points and adding new evaluations to the manuscript. This allows the readers to better judge the relevance of the results. 

Reviewer 2 Report

The authors have answer all  this reviewers concerns.